# Nutritional Assessment in Inflammatory Bowel Disease (IBD)—Development of the Groningen IBD Nutritional Questionnaires (GINQ)

**DOI:** 10.3390/nu11112739

**Published:** 2019-11-12

**Authors:** Vera Peters, Behrooz Z Alizadeh, Jeanne HM de Vries, Gerard Dijkstra, Marjo JE Campmans-Kuijpers

**Affiliations:** 1Department of Gastroenterology and Hepatology, University Medical Center Groningen and University of Groningen, 9713 GZ Groningen, The Netherlands; b.z.alizadeh@umcg.nl (B.Z.A.); gerard.dijkstra@umcg.nl (G.D.); m.j.e.campmans-kuijpers@umcg.nl (M.J.C.-K.); 2Department of Epidemiology, University Medical Center Groningen and University of Groningen, 9713 GZ Groningen, The Netherlands; 3Division of Human Nutrition and Health, Wageningen University and Research, 6708 PB Wageningen, The Netherlands; jeanne.devries@wur.nl

**Keywords:** Food Frequency Questionnaire (FFQ), Inflammatory Bowel Disease (IBD), Groningen IBD Nutritional Questionnaires (GINQ), dietary intake, dietary assessment

## Abstract

Diet plays a key role in the complex etiology and treatment of inflammatory bowel disease (IBD). Most existing nutritional assessment tools neglect intake of important foods consumed or omitted specifically by IBD patients or incorporate non-Western dietary habits, making the development of appropriate dietary guidelines for (Western) IBD patients difficult. Hence, we developed a food frequency questionnaire (FFQ), the Groningen IBD Nutritional Questionnaires (GINQ-FFQ); suitable to assess dietary intake in IBD patients. To develop the GINQ-FFQ, multiple steps were taken, including: identification of IBD specific foods, a literature search, and evaluation of current dietary assessment methods. Expert views were collected and in collaboration with Wageningen University, division of Human Nutrition and Health, this semi-quantitative FFQ was developed using standard methods to obtain a valid questionnaire. Next, the GINQ-FFQ was digitized into a secure web-based environment which also embeds additional nutritional and IBD related questions. The GINQ-FFQ is an online self-administered FFQ evaluating dietary intake, taking the previous month as a reference period. It consists of 121 questions on 218 food items. This paper describes the design process of the GINQ-FFQ which assesses dietary intake especially (but not exclusively) in IBD patients. Validation of the GINQ-FFQ is needed and planned in the near future.

## 1. Introduction

Diet is known to play a key role in the complex etiology of inflammatory bowel disease (IBD); comprising Crohn’s Disease (CD) and Ulcerative colitis (UC) [1]. Patients with IBD tend to adapt “unguided” dietary habits; without the guidance of a dietician or physician [2,3]. By adapting their dietary habits, patients intend to attenuate or even eliminate their suffering from IBD symptoms. A circular process of continuous experimentations is usually performed to identify trigger foods. Nevertheless, it often results in negative coping strategies [4] and nutritional deficiencies (including iron, calcium, vitamin D, vitamin B12, folic acid, zin, magnesium, and vitamin A) [5,6] due to the lack of sufficient knowledge of patients. When patients do seek dietary advice, physicians and dieticians themselves are struggling to provide IBD patients with proper evidence-based nutrition guidance due to the lack of hard scientific evidence for relationships of foods with IBD, inconsistency among these findings, and inconclusiveness of results from previous studies [7].

Lack of consensus about beneficial and harmful nutrition in patients with IBD is partly due to studies being affected by methodological issues such as information bias, recall bias, lack of proper measurement instruments or statistical incompliance [8]. Additionally, dietary assessment tools are mostly designed and validated for the general population [9], and notably neglect nutrients and foods that patients with IBD consume extensively or avoid deliberately. This hampers the development of effective dietary guidelines for (western) IBD patients. Patients report to avoid or have higher intakes of certain products as coping strategies [10]. Since IBD patients often have these unguided dietary habits, an IBD specific tool to access their dietary intake is needed. Only when the intake of IBD patients is registered correctly, can nutritional gaps be identified. Identification of nutritional gaps is needed to facilitate the development of evidence-based dietary guidelines and subsequently, give correct dietary advice to IBD patients. To get a complete overview of the intake of these IBD patients (and their unguided habits), we aimed to develop a tool that registers food intake of these patients, includes food items of significant interest for IBD and specifies food habits that patients are likely to adapt due to their reflection on disease complaints.

To achieve qualitative dietary guidance, we took the first steps in a multistep approach. We composed a new food frequency questionnaire (FFQ), called the Groningen IBD Nutritional Questionnaires (GINQ-FFQ), dedicated (but not limited) to assessing dietary intake in IBD patients. Development of this new nutritional assessment tool could potentially catalyze the development of evidence-based nutritional therapeutic strategies since this tool may identify nutritional gaps. The aim of this present paper is to describe how this dietary assessment tool was developed.

## 2. Methods

To develop the GINQ-FFQ, several steps were taken (Figure 1). First, we identified important foods that are often consumed or omitted by IBD patients; through a literature search, interviewing patients for their personal experiences, and expert opinions. Secondly, we searched the literature to identify which FFQs are being used to assess dietary intake of IBD patients and evaluated whether and how these FFQs met our purpose. Thirdly, we compiled the GINQ-FFQ using a validated FFQ used in the Leiden Longevity Study (described elsewhere [11]) as a base-model. The GINQ-FFQ includes foods that are preferably consumed or avoided by IBD patients. Fourthly, the newly developed GINQ-FFQ was digitized together with additional questions and several already existing questionnaires on nutrition and IBD related topics. Fifthly, the questionnaire needs to be validated. All steps are explained in more detail below.

### 2.1. Step 1—Identification of IBD Specific Foods

Our goal is to design an FFQ specific for IBD patients. Therefore, as initial step, the literature was searched to identify IBD specific foods, meaning foods that have been reported to be important in the development or treatment of IBD or are preferably consumed or intentionally avoided by IBD patients as a coping strategy for their disease. This procedure has led us to form a comprehensive list of food items which build the backbone of the GINQ-FFQ.

### 2.2. Step 2—Evaluation of Current Methods

Secondly, we conducted a literature search to identify what FFQs are currently used to assess dietary intake of IBD patients worldwide. The following search terms were used to search the PubMed database: “Food frequency questionnaire” AND “Inflammatory Bowel Disease (IBD)”. We compared the received questionnaires reciprocally and distilled important (IBD-) foods and added these to the list of our previously identified foods (step 1).

### 2.3. Step 3—Composition of the GINQ-FFQ

Foods identified in step 1 were selected to be part of the GINQ-FFQ if they were nutrient-rich (such as plant-based milk alternatives), consumed reasonably often (e.g., nuts, vegetable, juices), or attributed to between-person variations (e.g., fast food). Moreover, we focused on foods that were consumed or excluded specifically by IBD patients [12], or foods that are known to be consumed differently between patients and controls (i.e., compared to controls; less potential beneficial foods and more potential deleterious foods are consumed by patients [3]). The selection of these items was made based on expert views from physicians and dieticians. All selected food items were grouped into breakfast, lunch, dinner, snacks, or beverages to avoid major missing items. Then, for the design of the GING-FFQ, we collaborated with Wageningen University and Research (WUR), division of Human Nutrition and Health, due to their expertise on human nutrition. They use their FFQ Tool^™^ for realizing and validating web-based food frequency questionnaires (FFQ). This FFQ Tool^™^ converts frequency of food items into nutritional intake using the Dutch food composition database (NEVO table/“Nederlands Voedingsstoffenbestand”) [13]. Intake is calculated for 23 macronutrients (total protein, plant protein, animal protein, total fat, sum fatty acids, saturated fatty acids, monounsaturated fatty acids, polyunsaturated fatty acids, linoleic acid C18:2 cc, monounsaturated fatty acids cis, total trans fatty acids, n-3 polyunsaturated fatty acids cis, n-6 polyunsaturated fatty acids cis, alpha-linolenic acid C18:3(n-3) cis, eicosapentaenoic acid C20:5(n-3) cis, docosahexaenoic acid C22:6(n-3) cis, cholesterol, total carbohydrates, total mono/disaccharides, polysaccharides, dietary fiber, water, and alcohol) and 25 micronutrients (calcium, phosphor, total iron, iron haem, iron non-haem, sodium, potassium, magnesium, zinc, selenium, copper, iodide, retinol, retinol equivalent, vitamin B1, vitamin B2, vitamin B6, vitamin B12, vitamin D, vitamin E total, vitamin C, folic acid added, retinol activity equivalents (RAE), folic acids equivalents, and total organic acids). The previously selected food items were transformed into frequency questions via this FFQ Tool^™^ and added to the already existing base-model questionnaire (FFQ used in Leiden Longevity Study [11]). When there were already similar questions, those questions were modified according to our findings.

After formation of the GINQ-FFQ, additional expert views (physicians, dieticians, and researchers) were collected and several items were excluded from the questionnaire based on their views; when items did not add enough to between-person variation or to shorten the questionnaire to secure data quality. Questionnaire length is generally assumed to have an effect on survey response rate; respondents will get tired, bored and/or distracted by external factors and are therefore less likely to complete and return a lengthy questionnaire [14]. In addition, items that could not be linked to food composition data in the Dutch food composition database (NEVO table) [13] were not incorporated in the final GINQ-FFQ. Moreover, because of technicalities with the FFQ Tool^™^ a few more items could not be incorporated. Finally, a 218-item food frequency questionnaire was composed (the GINQ-FFQ). This questionnaire assesses the dietary intake of a patient, taking the preceding month as a reference period.

### 2.4. Step 4—Digitization of the GINQ

Collaboration with different parties was needed to transform the newly designed GINQ into a web-based questionnaire. The WUR transformed the GINQ-FFQ into an online questionnaire with use of their FFQ Tool^™^. Additional to the FFQ-part, several questions and questionnaires were digitized at the same time and added to the GINQ-FFQ. Therefore, we collaborated with Research Data Support (RDS) for their expertise in data collection. These additional items included questions on nutritional supplement use and the validated Food-Related Quality of Life questionnaire (FR-QoL) [15] to measure patients’ opinions on the effect of nutrition in IBD. Furthermore, general/demographic data, current health status, general health status, and IBD-related family history were incorporated. For the disease activity status, the recently validated Monitor IBD At Home (MIAH) questionnaire [16] was used. For physical activity the Baecke activity level questionnaire [17], [18] was incorporated. These items were selected based on expert views (physicians and researchers). RDS digitized and incorporated these additional questions and already existing questionnaires into the GINQ in Utopia (University Medical Center Groningen (UMCG), Groningen, The Netherlands); a UMCG developed application for supporting data processes for studies. Utopia was developed using Microsoft C# and all data is stored in Microsoft SQL Server. The two systems (FFQ Tool^™^ and Utopia) were linked to improving patient convenience when filling out the GINQ. All questionnaires together are called the GINQ. The readability of the GINQ was tested; the questionnaire was sent to participants (*n* = 14) and they were asked to give feedback on the GINQ. Changes were made where necessary. Additionally, participants were asked to register the time needed to fill out the GINQ.

### 2.5. Step 5—Validation of the GINQ-FFQ

The GINQ-FFQ is based on standard methods of WUR and therewith the content validity and face validity are assured. Construct validity, precision, and reproducibility still need to be checked; this will be done in a follow-up study. To validate the GINQ-FFQ, we plan to compare it to mean dietary intake deducted from 24-hour dietary recalls. Therefore, we will administer the GINQ-FFQ and additional three times 24h dietary recall to approximately 300 IBD patients within our outpatient IBD population.

## 3. Results

The first step in the process of developing the GINQ-FFQ was to compile a comprehensive list of food items based on the literature and information provided by patients, dieticians, and physicians. This list is presented in Appendix A: Identification of foods.

Secondly, we reached out to IBD-researchers who used an FFQ in their studies. As shown in the flowchart (Figure 2), 42 articles were found and eventually 30 authors could be contacted about their applied methods. After one month 21 authors did not respond, and we received positive responses from nine authors (Hekmat [19], Ananthakrishnan [20], Day [21], Shatenstein [22], El Mouzan [23], Khalili [24], Barrett [25], Lewis [26], and Chiba [27]). The received responses included six FFQs and one list of several short questions. Two researchers used the same questionnaire. Eventually, five FFQs were compared reciprocally. This comparison is presented in Appendix A: Comparison of FFQs. We concluded that dairy, bread and bread substitutes, meat and meat substitutes, fish, additives to meals (e.g., superfoods, spices, herbs), sweet snacks, and beverages should be questioned more extensively in the GINQ-FFQ than was done in most of these questionnaires. Additionally, out of these questionnaires, we selected and added important foods that were not yet on our extended list of Step 1.

Thirdly, based on expert views (physicians, dieticians, and researchers), several items from the extensive list composed in step 1 and step 2 were excluded from being incorporated in the GINQ-FFQ. Reasons for exclusion were: (1) on several items, no food composition data was available in the Dutch food composition database (NEVO table [13]) (e.g., sushi and naan bread), (2) products that did not add to the between-person variation (e.g., olives) were not included and (3) to shorten the questionnaire some items were not included or summarized (e.g., eggs used in dishes or eaten with bread were summarized in one question); meaning products that were not consumed regularly enough or did not attribute enough to the total intake.

In step 4 we digitized the questionnaire. Due to some technicalities with the FFQ Tool^™^, it was impossible to calculate the composition of some foods (e.g., seasoning, fast food, etc.) and subsequently these items were not incorporated. After expert views and technical implementation, 218 items ended up in the FFQ. To get some insight in the level of fresh food vs. ultra-processed foods consumed by patients, fast food products were incorporated in additional questions. The time needed to fill out the questionnaire was reported by participants during the readability process. It took them approximately 45 min to fill out the GINQ-FFQ. Additional nutritional and IBD related items were digitized as well. The result was a patient friendly online GINQ which contains, in total, 204 questions (web-version); 121 food frequency questions on 218 food items and embeds 83 questions on additional nutritional and IBD related topics.

## 4. Discussions

A new IBD-specific FFQ—The GINQ-FFQ—was developed (and digitized) to assess dietary intake of IBD patients, using the preceding month as a reference period. The result is an FFQ with 121 questions on 218 food items embedded in an online web environment together with 83 additional questions on nutrition and IBD related topics. It takes approximately 45 min to fill out the GINQ-FFQ. Where other FFQs neglect the intake/avoidance of IBD specific products and Western dietary habits, the GINQ-FFQ includes as many as possible of these foods. The GINQ-FFQ assesses dietary intakes of IBD patients in an extensive way, which is urgently needed for the development of proper dietary guidelines and dietary advice for patients.

Food-related coping strategies are common among IBD patients; Vagianos et al. [12] found that patients, especially those with active IBD, more often avoid products like alcohol, popcorn, legumes, nuts, seeds, deep fried foods, and processed deli meat, while consuming more sports drinks and sweetened beverages. As reported by De Vries et al. [10], patients often report avoiding spicy foods, carbonated drinks, milk and dairy products, energy drinks, deep-fried foods, alcoholic drinks, cabbages, pork, processed meat, coffee, pastries, sweets, citrus fruits, low-fiber bread, legumes, and peanuts but state taking in more high-fiber bread, tea, leafy vegetables, fatty fish, poultry, exotic fruits, pit fruits, soft fruits, non-leafy vegetables, eggs, legumes, wholegrain rice, nuts, white fish, and cabbage. Jowett et al. [28] found that patients with UC avoided the following food groups: milk and dairy products, fruit and vegetables, meat, fish and alternatives, bread, other cereals and potatoes, fatty and sugary foods, spicy foods, legumes, alcohol, and high fiber foods. They also found that some food groups were reported to be eaten in greater amounts because patients felt eating these products was helping to ameliorate symptoms: milk and dairy foods, fruits and vegetables, meat, fish and alternatives, bread, other cereals and potatoes, fatty and sugary foods, spicy foods, and high fiber foods. The fact that these studies on patients’ dietary beliefs [10,28,29,30,31,32] report different results attributed to the idea that it is important to register the intake of such items in our GINQ-FFQ; products that are actively avoided or consumed extensively by IBD patients as dietary strategies to extenuate IBD symptoms. By incorporating these foods (including questions on (raw) vegetables, (dried) fruits, fast food, nuts, dairy (yoghurt specifically) and dairy alternatives, meat alternatives, super foods, herbs and spices, and chewing gum) in the GINQ-FFQ, we achieved a complete overview of patients’ dietary habits and covered the identified gaps in registration of their nutritional intake (e.g., to improve nutritional deficits).

To obtain long-term dietary intake, using an FFQ is the method of choice and is recommended by nutritional epidemiologists [14]. Using an FFQ to assess dietary intake always comes with general limitations because an FFQ is based on a predefined list with food items and detailed information about food preparation and brands, and contextual information about intake (e.g., which items are consumed at the same meal) is lacking. In addition, it is known that quality may decline if an FFQ includes too many questions [14]. Moreover, this method is not suitable to collect information on the degree of how (ultra-) processed foods are and whether patients use many ready-made products to “prepare” their meals. On the other hand, it is commonly known that an FFQ is a proper tool to assess dietary intake over a longer period of time. Additionally, this tool is inexpensive and can be used in population-based large epidemiology studies. This explains why FFQs are currently one of the most widely used dietary assessment tools in the field of nutritional epidemiology [14].

The strength of the GINQ-FFQ is that the predefined list on which the FFQ-part is based, is extensive and includes items that are specific for the IBD population. Therefore, it allows us to get a good overview of the intake of IBD patients, even if they have these particular unguided habits. Since the GINQ-FFQ covers a broad range of food items and was composed in collaboration with WUR using their standard methods to create a valid questionnaire, the use of this FFQ is designed for IBD patients but is not limited to this specific population. This newly developed questionnaire may also be used in populations with immune-mediated inflammatory diseases (IMIDs) [33] or in healthy controls. This tool was designed to be used in different institutes to facilitate the comparison of results between these institutes. Additionally, the GINQ will be incorporated into “MyIBDCoach” [34], an app used to monitor patients to enhance patient awareness and to improve patient empowerment by giving direct feedback on their dietary intake. Moreover, the use of a web-based environment increases convenience for patients and can be utilized and combined with other IBD related nutrition questionnaires.

Specific limitations of the GINQ-FFQ are that certain items of interest specific for IBD could not be included due to lack of information on the food composition, technicalities with the FFQ Tool^™^, and to assure the questionnaire is not too extensive. Since content and face validity are already assured due to the standard methods of WUR, only the construct validity, precision, and reproducibility of the GINQ-FFQ need to be checked and this will be done in a follow-up study.

## 5. Conclusions

The GINQ is potentially an appropriate dietary assessment tool for future epidemiological studies in Western IBD populations. Although validity and reliability of this newly developed questionnaire is planned to be studied in the near future, development of this GINQ-FFQ is a first necessary step in increasing our knowledge on the role of diet in IBD and to guide personalized anti-inflammatory dietary advice in the future.

## Figures and Tables

**Figure 1 nutrients-11-02739-f001:**
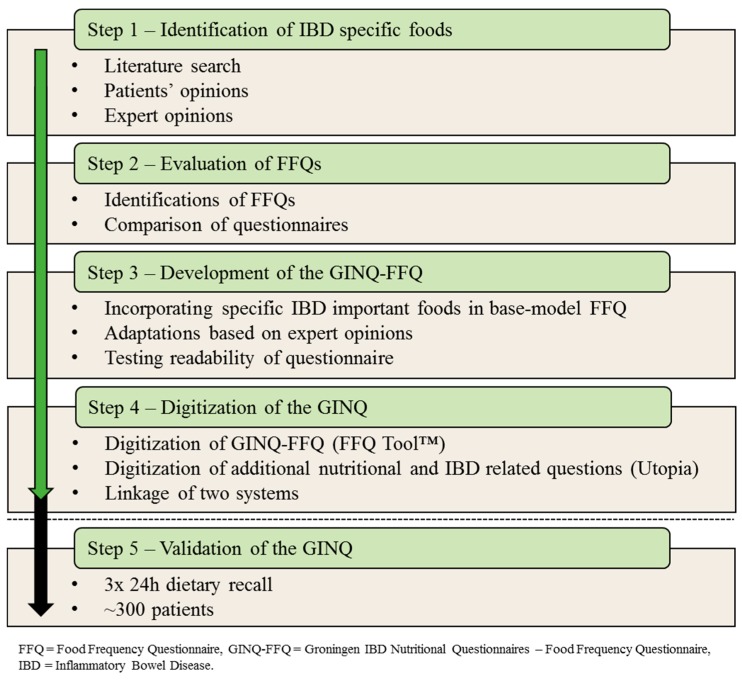
Flowchart development of the GINQ.

**Figure 2 nutrients-11-02739-f002:**
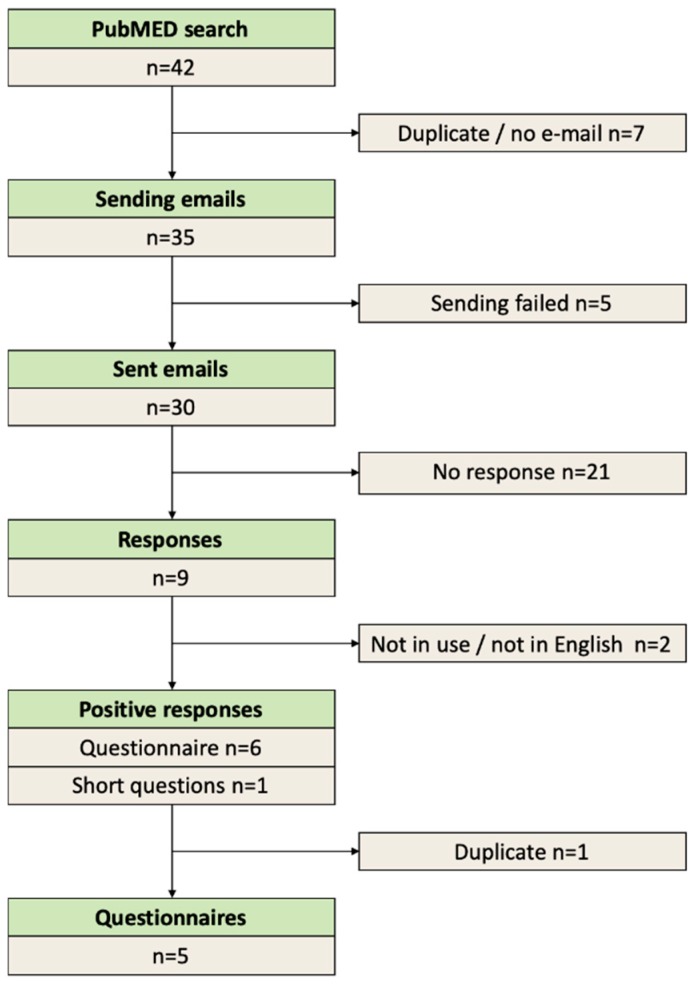
Flowchart identifying FFQs.

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
