# Peer review of "Nutritional Assessment in Inflammatory Bowel Disease (IBD)—Development of the Groningen IBD Nutritional Questionnaires (GINQ)"

_nutrients, 2019, doi:10.3390/nu11112739_

Round 1
Reviewer 1 Report
Dr. Peters and colleagues described the design process of the GINQ-FFQ, a newly designed tool to assess dietary intake, particularly in IBD patients.
The topic is interesting, the manuscript is well written and presented. I have some comments:
in the Results section, the authors state that it takes approx. 45 minutes to fill out the questionnaire. However, in the Methods section they did not refer to the measuring process. the authors state that the dietary intake in the preceding month is a proxy for their habitual intake over the last six months. Is it a speculation or do they refer to a specific reference? the authors conclude that the GINQ-FFQ is an appropriate tool for future epidemiological studies. As they discussed in the limitations, this statement is still not supported by the study.Author Response
Rebuttal – Comments from reviewers on manuscript: “Nutritional assessment in Inflammatory Bowel Disease (IBD)—development of the Groningen IBD Nutritional Questionnaires (GINQ)” – V. Peters et al.
All references to the revised manuscript (“line(s) xxx”) refer to the revised manuscript file.
General reply from the authors
First of all, we would like to thank the reviewers for their kind words and for their thorough assessment of our manuscript. This motivated us to further improve our manuscript according to the raised comments and suggestions. Please find below our detailed point-by-point response to all individual reviewers’ comments (written in blue). Besides, we have proofread the manuscript again and made changes were necessary (see revised manuscript – marked yellow lines).
Comments and suggestions reviewer #1
“Dr. Peters and colleagues described the design process of the GINQ-FFQ, a newly designed tool to assess dietary intake, particularly in IBD patients. The topic is interesting, the manuscript is well written and presented.” Thank you for these compliments.
“I have some comments: in the result section, the authors state that it takes approx. 45 minutes to fill out the questionnaire. However, in the methods section they did not refer to the measuring process.” We agree with the reviewer that we could have described our measuring process in more detail. Therefore, we added more details about the procedures in the method section: “The readability of the GINQ was tested; the questionnaire was sent to participants (n=14) and they were asked to give feedback on the GINQ. Changes were made where necessary. Besides, participants were asked to register the time needed to fill out the GINQ(-FFQ).” (lines 136-139) and in the results section: “Time needed to fill out the questionnaire was reported by participants during the readability process. It took them approximately 45 minutes to fill out the GINQ-FFQ.” (lines 174-176).
“The authors state that the dietary intake in the preceding month is a proxy for their habitual intake over the last six months. Is it a speculation or do they refer to a specific reference?” This statement is not purely based on speculation as suggested. However, it was based on internal communication, but currently no reference is available to support this statement. In the GINQ-FFQ we asked people about their intake over the past months. Therefore, we adapted our manuscript: “The GINQ-FFQ is an online self-administered FFQ evaluating dietary intake, taking the past month as a reference period.” (lines 23-24), “This questionnaire assesses the dietary intake of a patient, taking the preceding month as a reference period.” (lines 120-121) and “A new IBD-specific FFQ – the GINQ-FFQ – was developed (and digitalized) to assess dietary intake of IBD patients, using the preceding month as reference period.” (lines 183-184).
“The authors conclude that the GINQ-FFQ is an appropriate tool for future epidemiological studies. As they discussed in the limitations, this statement is still not supported by the study.” Indeed, this might be formulated too firmly. We changed the sentence into “The GINQ is potentially an appropriate dietary assessment tool […]”(Line 240-241) and “Development of this new nutritional assessment tool could potentially catalyze the development of evidence-based nutritional therapeutic strategies since this tool may identify nutritional gaps.” (lines 59-60). Besides, to clarify how we plan to go from development of a questionnaire to the use of it in epidemiological studies in the future, we added an additional statement and step in the method section: “Fifthly, the questionnaire needs to be validated.” (line 71), “Step 5 – Validation of the GINQ. The GINQ-FFQ is based on standard methods of WUR and therewith content validity and face validity is assured. Construct validity, precision and reproducibility still need to be checked; this will be done in a follow-up study. To validate the GINQ-FFQ, we plan to compare it to mean dietary intake deducted from 24-hour dietary recalls. Therefore, we will administer the GINQ-FFQ and additionally three times a 24h dietary recall to approximately 300 IBD patients within our outpatient IBD population.” (lines 141-146) and adapted figure 1 accordingly (lines 73-74). Besides, we expressed it in the abstract more clearly: “Validation of the GINQ-FFQ is needed and planned in the near future.” (lines 26-27) and adapted our statement in the discussion: “Since content and face validity are already assured due to the standard methods of WUR, only the construct validity, precision and reproducibility of the GINQ-FFQ need to be checked and this will be done in a follow-up study.” (lines 237-239).
Given the limitations of other questionnaires to assess dietary gaps in the intake of IBD patients, we do feel that an appropriate questionnaire for this specific population is needed. When comparing the FFQ’s in step 2, we found that (supplementary file 2) dairy, bread and bread substitutes, meat and meat substitutes, fish, additives to meals (e.g. superfoods, spices, herbs), sweet snacks and beverages should be questioned more extensively in the GINQ-FFQ than was done in most of these questionnaires (lines 157-160). Additionally, we decided to go into more detail when questioning food items (supplementary file 1) such as (raw) vegetables, (dried) fruits, fast food, nuts, yoghurt, milk alternatives, meat alternatives, super foods, herbs and spices, chewing gum. We adapted this in the discussion: “By incorporating these foods (including questions on (raw) vegetables, (dried) fruits, fast food, nuts, dairy (yoghurt specifically) and dairy alternatives, meat alternatives, super foods, herbs and spices, chewing gum) in the GINQ-FFQ, [...]” (lines 207-210).

Reviewer 2 Report
The authors have provided an extensive description of how they have developed a FFQ questionnaire by incorporating food-related questions following literature review, expert opinions, adapting questions from existing questionnaires, collaborating with University faculties with expertise on nutrition and finally transforming it into a digitalised web-based format. The paper focuses purely on the description of methods used to develop the questionnaire. The authors state that the importance of this questionnaire is to develop future dietary guidelines and accordingly formulate dietary advice to patients with IBD. This is however not a validated questionnaire and therefore the wide readership would be less inclined to implement it at this stage. It is unclear from the paper what steps would follow after the questionnaire is filled out, how the gaps in nutrition would be identified, how and who would synthesise the information and how that would result in the development of nutritional guidelines. It is unclear how the FFQ is more suited to patients with IBD; what are the ‘peculiar’ dietary habits in patients with IBD; what foods are they more likely to avoid and therefore result in nutritional deficiencies. I understand that the Questionnaire aims to do just that. However, its important to provide this information based on currently available literature. As this paper is descriptive focusing on the methodology rather than findings following implementation of the questionnaire, I am struggling to accept the scientific value of this paper.
Author Response
Rebuttal – Comments from reviewers on manuscript: “Nutritional assessment in Inflammatory Bowel Disease (IBD)—development of the Groningen IBD Nutritional Questionnaires (GINQ)” – V. Peters et al.
All references to the revised manuscript (“line(s) xxx”) refer to the revised manuscript file.
General reply from the authors
First of all, we would like to thank the reviewers for their kind words and for their thorough assessment of our manuscript. This motivated us to further improve our manuscript according to the raised comments and suggestions. Please find below our detailed point-by-point response to all individual reviewers’ comments (written in blue). Besides, we have proofread the manuscript again and made changes were necessary (see revised manuscript – marked yellow lines).
Comments and suggestions reviewer #2
“The authors have provided an extensive description of how they have developed an FFQ by incorporating food-related questions following literature review, expert opinions, adapting questions from existing questionnaires, collaborating with university faculties with expertise on nutrition and finally transforming it into a digitalized web-based format. The paper focuses purely on the description of methods used to develop the questionnaire. The authors state the importance of this questionnaire is to develop future dietary guidelines and accordingly formulate dietary advice to patients with IBD. This is however not a validated questionnaire and therefore the wide readership would be less inclined to implement it at this stage.” This is indeed correct since the validation of the questionnaire is the next step in the described multistep process. Currently, we are setting up this validation study. To emphasize that the questionnaire is not yet validated, we stated this more explicitly in the abstract: “Validation of the GINQ-FFQ is needed and planned in the near future.” (lines 26-27). Besides, we added an additional statement and step in the method section: “Fifthly, the questionnaire needs to be validated.” (line 71), “Step 5 – Validation of the GINQ. The GINQ-FFQ is based on standard methods of WUR and therewith content validity and face validity is assured. Construct validity, precision and reproducibility still need to be checked; this will be done in a follow-up study. To validate the GINQ-FFQ, we plan to compare it to mean dietary intake deducted from 24-hour dietary recalls. Therefore, we will administer the GINQ-FFQ and additionally three times a 24h dietary recall to approximately 300 IBD patients within our outpatient IBD population.” (lines 141-146), and adapted figure 1 accordingly (lines 73-73). Next, we adapted our statement in the discussion: “Since content and face validity are already assured due to the standard methods of WUR, only the construct validity, precision and reproducibility of the GINQ-FFQ need to be checked and this will be done in a follow-up study.” (lines 237-239).
“It is unclear from the paper what steps would follow after the questionnaire is filled out, how the gaps in nutrition would be identified, how and who would synthesize the information and how that would result in the development of nutritional guidelines.” Knowing the real dietary intake is a prerequisite to be able to develop dietary guidelines; proper tools for this dietary assessment are needed. Therefore, being able to measure their valid dietary intake via an IBD-specific FFQ is a necessary first step. Then, the GINQ-FFQ needs to be validated before implementation (adapted as discussed before). When the questionnaire is implemented, it can capture a more complete overview than other questionnaire because it contains more detailed information on certain food groups and food items, including more IBD specific items (supplementary 1); e.g. food groups: dairy, bread and bread substitutes, meat and meat substitutes, fish, additives to meals (e.g. superfoods, spices, herbs), sweet snacks and beverages, and food items: (raw) vegetables, (dried) fruits, fast food, nuts, yoghurt, milk alternatives, meat alternatives, super foods, herbs and spices and chewing gum. We adapted this in the discussion: “By incorporating these foods (including questions on (raw) vegetables, (dried) fruits, fast food, nuts, dairy (yoghurt specifically) and dairy alternatives, meat alternatives, super foods, herbs and spices, chewing gum) in the GINQ-FFQ, [...]” (lines 207-210). Next, the implementation of this questionnaire in the IBD population may lead to the identification of nutritional gaps (since it was developed for this population and captures 23 macro- and 25 micronutrients (lines 99-108)) and can therewith potentially catalyze the development of dietary measures and guidelines for IBD patients. However, (how to come to) the development of dietary guidelines is not within the scope of this manuscript and should be based on many studies. Therefore, we did not describe the future steps that should be based on a jointed effort within and between the nutrition and gastroenterology field(s).
“It is unclear how the FFQ is more suited to patients with IBD; what are the “peculiar” dietary habits in patients with IBD; what foods are they more likely to avoid and therefore result in nutritional deficiencies. I understand that the Questionnaire aims to do just that. However, it is important to provide this information based on currently available literature. Nevertheless, it is important to provide information on food habits of IBD patients to the readers.” As was stated in our manuscript in lines 52-55 and 76, the goal of this article is to design an FFQ specific for IBD patients. To state this even more clearly, we adapted the following: “Patients report to avoid or have higher intakes of certain products as coping strategy.[1] Since IBD patients often have these unguided dietary habits, an IBD specific tool to access their dietary intake is essentially needed. Only when intake of IBD patients is registered correctly, nutritional gaps can be identified. Identification of nutritional gaps is needed to facilitate the development of evidence-based dietary guidelines and subsequently, giving correct dietary advice to IBD patients. To get a complete overview of the intake of these IBD patients (and their unguided habits), we aimed to develop a tool that registers food intake of these patients, includes food items of significant interest for IBD and specifies food habits which patients are likely to adapt due to their reflection on disease complaints.”(lines 47-55).
In the discussion we explained why the GINQ-FFQ is more suited for IBD patients (e.g. includes food items that patients avoid or gourmandize according to literature) but is not limited to this specific population (lines 25-26, 52-55, 186-190, 223-228, supplementary table 1 and 2). Nevertheless, we tried to improve our explanation on why it is important to develop such tools for IBD patients, so we adjusted our method section “Our goal is to design an FFQ specific for IBD patients. Therefore, as initial step, literature was searched to identify IBD specific foods, meaning foods that have been reported to be important in the development or treatment of IBD or are preferably consumed or intentionally avoided by IBD patients as a coping strategy for their disease.” (lines 76-79).
Patients are more at risk for nutritional factor deficiencies; including iron, calcium, vitamin D, vitamin B12, folic acid, zin, magnesium and vitamin A. This increased risk is due to chronic inflammation as well as side effects of chronic use of medications.[2] We adapted this additional information: “[…] nutritional deficiencies (including iron, calcium, vitamin D, vitamin B12, folic acid, zin, magnesium and vitamin A), […]” (lines 36-38).
Thank you for pointing out that we did not state clearly enough what patients often over consume or avoid. We think the reviewer made a legit point and therefore, we added more currently available literature: “As reported by De Vries et al[1], patients often report to avoid spicy foods, carbonated drinks, milk and dairy products, energy drinks, deep-fried foods, alcoholic drinks, cabbages, pork, processed meat, coffee, pastries, sweets, citrus fruits, low-fiber bread, legumes and peanuts but state to take in more high-fiber bread, tea, leafy vegetables, fatty fish, poultry, exotic fruits, pit fruits, soft fruits, non-leafy vegetables, eggs, legumes, wholegrain rice, nuts, white fish and cabbage. Jowett et al[3] found that patients with UC avoided the following food groups: milk and dairy products, fruit and vegetables, meat, fish and alternatives, bread, other cereals and potatoes, fatty and sugary foods, spicy foods, legumes, alcohol and high fiber foods. They also found that some food groups were reported to be eaten in greater amount because patients felt eating these products was helping to ameliorate symptoms: milk and dairy foods, fruits and vegetables, meat, fish and alternatives, bread, other cereals and potatoes, fatty and sugary foods, spicy foods, high fiber foods.” (lines 194-204).
“As this paper is descriptive focusing on the methodology rather than findings following implementation of the questionnaire, I am struggling to accept the scientific value of this paper.” This article points out the necessity to have a special FFQ for IBD patients as they are eating special foods and omitting typical foods that cannot be captured in existing questionnaires. Therefore, we searched in literature which current questionnaires are being used and asked authors to provide these questionnaires. We compared these questionnaires and all dietary items, but especially those concerning dietary habits in IBD patients. On this basis we developed a new questionnaire, suitable to IBD patients but also to other patients. Although the GINQ-FFQ still needs validation, we do think this GINQ-FFQ is of scientific value to come to proper results of dietary intake in especially IBD patients, which is a first step to come to dietary guidelines.
References
[1] J. H. M. de Vries, M. Dijkhuizen, P. Tap, and B. J. M. Witteman, “Patient’s Dietary Beliefs and Behaviours in Inflammatory Bowel Disease.,” Dig. Dis., vol. 37, no. 2, pp. 131–139, 2019.
[2] D. Owczarek, T. Rodacki, R. Domagała-Rodacka, D. Cibor, and T. Mach, “Diet and nutritional factors in inflammatory bowel diseases.,” World J. Gastroenterol., vol. 22, no. 3, pp. 895–905, Jan. 2016.
[3] S. L. Jowett, C. J. Seal, E. Phillips, W. Gregory, J. R. Barton, and M. R. Welfare, “Dietary beliefs of people with ulcerative colitis and their effect on relapse and nutrient intake,” Clin. Nutr., vol. 23, no. 2, pp. 161–170, Apr. 2004.

Reviewer 3 Report
I read with interest this relevant MS about developing a nutritional score for IBD patients. It a well developed study. I have a minor suggestion: On the clinical side would be of help to have some correlation of diet with some IBD score and/or disease activity condition. No other suggestions on this side.
Author Response
Rebuttal – Comments from reviewers on manuscript: “Nutritional assessment in Inflammatory Bowel Disease (IBD)—development of the Groningen IBD Nutritional Questionnaires (GINQ)” – V. Peters et al.
All references to the revised manuscript (“line(s) xxx”) refer to the revised manuscript file.
General reply from the authors
First of all, we would like to thank the reviewers for their kind words and for their thorough assessment of our manuscript. This motivated us to further improve our manuscript according to the raised comments and suggestions. Please find below our detailed point-by-point response to all individual reviewers’ comments (written in blue). Besides, we have proofread the manuscript again and made changes were necessary (see revised manuscript – marked yellow lines).
Comments and suggestions reviewer #3
“I read with interest this relevant manuscript about developing a nutritional score for IBD patients. It is a well-developed study.” Thank you for this compliment.
“I have a minor suggestion: on the clinical side would be of help to have some correlation of diet with some IBD score and/or disease activity condition. No other suggestions on this side.” Thank you for this comment. In fact we already incorporated a disease activity score in our GINQ (part of the nutrition- and IBD-related additional questions/questionnaires), namely the Measure IBD at home (MIAH)[1] score (lines 130-131). The MIAH was recently validated and is the first patient reported outcome measure (PROM) which can accurately predict endoscopic inflammation in IBD patients when it is combined with a fecal calprotectin home test. Additionally, when the dietary intake is captured and calculated by our GINQ-FFQ, indices (e.g. Dutch Healthy Diet Index[2]) can be used to calculate a diet score and can then be related to the disease activity scores such as the MIAH.
References
[1] M. J. de Jong et al., “Development and Validation of a Patient-reported Score to Screen for Mucosal Inflammation in Inflammatory Bowel Disease,” J. Crohns. Colitis, vol. 13, no. 5, pp. 555–563, Apr. 2019.
[2] L. van Lee, E. J. M. Feskens, E. J. C. Hooft van Huysduynen, J. H. M. de Vries, P. van ’t Veer, and A. Geelen, “The Dutch Healthy Diet index as assessed by 24 h recalls and FFQ: associations with biomarkers from a cross-sectional study.,” J. Nutr. Sci., vol. 2, p. e40, 2013.

Round 2
Reviewer 2 Report
The authors have adequately addressed the questions I raised following my initial review.